# The shaping of social and symbolic capital during the transition to farming in the Western Mediterranean: Archaeological network analyses of pottery decorations and personal ornaments

Daniel Pereira[1]*, Claire Manen[2], Solange Rigaud[1]

**1** CNRS, Université de Bordeaux, Ministère de la Culture, PACEA, UMR 5199, Pessac, France, **2** CNRS UMR5608 TRACES. Université de Toulouse Jean-Jaurès, Maison de la Recherche, Toulouse, France

* daniel.da-silva-pereira@u-bordeaux.fr

**Data Availability Statement:** All relevant data are within the manuscript and its Supporting information files.

## Abstract

Storing information and circulating it between individuals and groups is a critical behaviour that signals a tipping point in our evolutionary history. Such practices enabled the preservation and consolidation of knowledge over extended periods, facilitating the accumulation of cultural innovations across generations. In this study, we used Social Network Analysis methods to explore how knowledge circulated during the transition to agriculture in the Western Mediterranean region. Previous studies have shown that specific elements of the material culture reveal distinct patterns of cultural interaction among early farming communities. Here, we investigated if two archaeological proxies, personal ornaments and pottery decorations, both with an exclusively symbolic function, reveal different network structures, and if the different degree of connexions acted equally on the transmission of styles, symbols, and network changes over time. Our results relied on cultural data recorded from 77 archaeological occupations covering Italy, France, and Spain, spanning over 1,500 years (ca. 7950~6450 cal BP). By utilizing a chronological dataset comprising 114 radiocarbon dates, we revealed that pottery decorative techniques networks exhibited stronger connexions over space and time, with nodes organized in clear cluster, when compared to personal ornaments networks. The findings highlight the regionalization and fragmentation of cultural networks during the Early Neolithic, and that the transmission of cultural traits within each category of artefact operated through varying cultural and social mechanisms. Pottery expressed a dynamic regional identity, continuously shaped by geographical and chronological proximity, while bead-type associations contributed to enduring identities shared across vast geographical scales. These networks shed light on the multifaceted shaping of social and symbolic capital among the Mediterranean's early farmers, emphasizing the strength and quality of social ties that existed between communities and the level of reciprocity and cooperation required to foster these diverse social, economic, and cultural development strategies.

**Funding:** The funders had no role in study design, data collection and analysis, decision to publish, or preparation of the manuscript. This work was supported by the French National Research Agency under the IDEX Bordeaux NETAWA Emergence project ANR-10-IDEX-03-02 'Out of the Core: Exploring social NETworks at the dawn of Agriculture in Western Asia 10 000 years ago' [SR], the CNRS Momentum project 'Symboling and neighboring at the dawn of Agriculture' [SR] and the Grand Programme de Recherche 'Human Past' of the University of Bordeaux [SR].

**Competing interests:** The authors have declared that no competing interests exist.

## Introduction

A key characteristic unique to the human lineage is our ability to materialize and store information outside the brain. Recently, this external storage of information has taken the form of writing, mass media, and the internet [1]. In the archaeological record, mobile and rock art, and personal ornaments–some of the material productions that attest to such an ability, whatever the symbolic message behind those productions–are attested as far back as 140 ky ago [2–7]. Our ability to store information and circulate it between individuals and groups is a critical behaviour that signals a tipping point in our evolutionary history. By doing so, we have been able to maintain and consolidate knowledge and cumulate cultural innovations over time [8–10]. But how did information and knowledge circulate during the past? How did past people structure their networks? And what mechanisms contributed to this network shaping?

The transfer of cultural traits within and between past communities has long been investigated. Contact between populations can be seen through the circulation of raw materials from their source to regions located several hundred kilometers away [11–15]. The sharing of common stylistic traits in pottery design [16–20], bone tool shaping and decoration [21,22], as well as flint weaponry production [23–25], are also commonly used to track interactions between communities. Maps describing connexions, circulation routes, and exchange networks have been produced for many periods of prehistory across many regions [26–28].

For the last three decades, Social Network Analysis (SNA) has been regularly used within various geographical and chronological contexts [29,30], especially over the past ten years, to explore various scenarios, such as the impact of demographic events on network structures [31,32], the mechanisms responsible for their long persistence [33], the social fabric of early villages [34], and multigenerational changes in network structure [35]. Here, we applied SNA to explore how pottery decorations and personal ornaments can be used as proxies for tracking the production, transmission, and accumulation of social [36,37] and symbolic capital during the transition to farming, whilst also emphasizing the strength and quality of social ties that existed between communities and their level of reciprocity and cooperation, all of which may have aided the successful diffusion of farming technologies throughout Europe.

This period represents the process by which human groups switched from hunting and gathering wild resources to a reliance on systems of food production based on domesticated plants and animals. In the Fertile Crescent, sedentism, farming, and herding progressively took place 15,000/12,000 years (y) ago, then spread across Europe from 8,800 until 5,500 y ago [38–40]. Increasingly refined archaeological [41–43], anthropological [44–47], and chronological data [48] have identified a succession of profound cultural, technical, and economic changes between the last indigenous hunter-gatherers and the first Early Neolithic farmers in Europe. This revolution played a pivotal role in establishing the economic and social underpinnings upon which many present-day societies rely, including diverse techniques for food production and storage, the emergence of surpluses, the shift towards sedentism, the specialization of labor, the growth of social complexity, and, ultimately, the formation of state institutions.

Previous studies have shown that a cultural substrate made up of ultra-connected European foraging communities may have represented favorable conditions for enhancing the rapid dispersion of the Neolithic way of life in Europe [28,49,50]. The vast networks documented between farming communities, and between local foraging communities and farmers, indicate that maintaining and reinforcing connexions between neighboring communities represented an efficient strategy for emerging farming societies seeking to spread and access new territory [49].

Our primary goal is to document the interactions and contact networks during the Neolithic transition in the Western Mediterranean region, in order to observe how past cultural diversity was shaped by repeated interactions and how it may have changed through time. We

postulate that the network structure between archaeological occupations may have impacted past mobility and the diffusion of knowledge, information, and innovations during the Early Neolithic.

## Research objectives, material, and methods

### Research objectives

Previous studies have shown that during the Early Neolithic human communities were highly connected and mobile, with well-developed large-scale circulation networks [51–54] that favoured both cultural and demic diffusions. Additional studies have evidenced that each mechanism acted to a different extent across Europe [55,56], although these interactions did not uniformly affect the various elements of the first farmers' material culture. Nevertheless, by comparing two proxies with an exclusively symbolic function, pottery decorations and personal ornaments, distinct patterns of interaction among early farming communities can be identified. Pottery decoration diversity indicates local circulation and exchange, resulting in the emergence and persistence of stylistic and symbolic distinctions between groups. In contrast, personal ornaments reflect extensive networks and the mobility of Early Neolithic farmers [57]. This previous comparative study did not, however, specifically explore the shape, structure, and density of the networks, preventing an exploration of how information flowed between groups. By applying various SNA methods, and by specifically applying this approach to both pottery decorative techniques and personal ornaments, we propose to investigate if the two archaeological proxies can reveal different network structures, if the communities who spread to the Western Mediterranean area developed tight or loose inter-connexions, if those connexions acted similarly on the transmissions of personal ornaments and pottery styles, and if networks were reshaped through time.

By assessing the extent of shared cultural traits among archaeological occupations, we can infer the degree of connexion between them, based on the premise that a greater cultural resemblance indicates stronger past interactions between communities. Our objective in exploring these networks of interactions is to identify communities of practice that shared common social and symbolic engagement.

We explored these questions using a large chronological scope covering the 1,500 years of Neolithic diffusion in the region through a time-sequential perspective. We explored how networks of interactions were reshaped through time, the potential lagged effects on the networks as communities moved to new locations, and the outcome of different communities separating or coalescing while spreading to new territories.

### Material

Our sample is constituted of two updated previously published datasets (for detailed information see Rigaud et al. 2018, Supplementary Material text B), including cultural and chronological data recorded from 77 archaeological occupations covering Italy, France, and Spain, spanning over 1,500 years (ca. 7950~6450 cal BP; Fig 1; worksheets A and C in S1 Dataset) [57]. The chronological dataset derives from 114 radiocarbon dates (worksheet J in S1 Dataset) and the cultural datasets from a typological classification system that we developed for the study of pottery decorative techniques [19,20,58] and personal ornaments [50,59].

The pottery dataset included 11 pottery decoration traits recorded within 44 Early Neolithic occupations (worksheet A, B and F in S1 Dataset). Because pottery fragment recovery is not as sensitive to field methods as beads, each variable was counted in each occupation included in the database. The visual aspect of decorative attributes may vary depending on the tools used to decorate the artefacts, the gestures used to apply the tools to the surface of the pottery, and

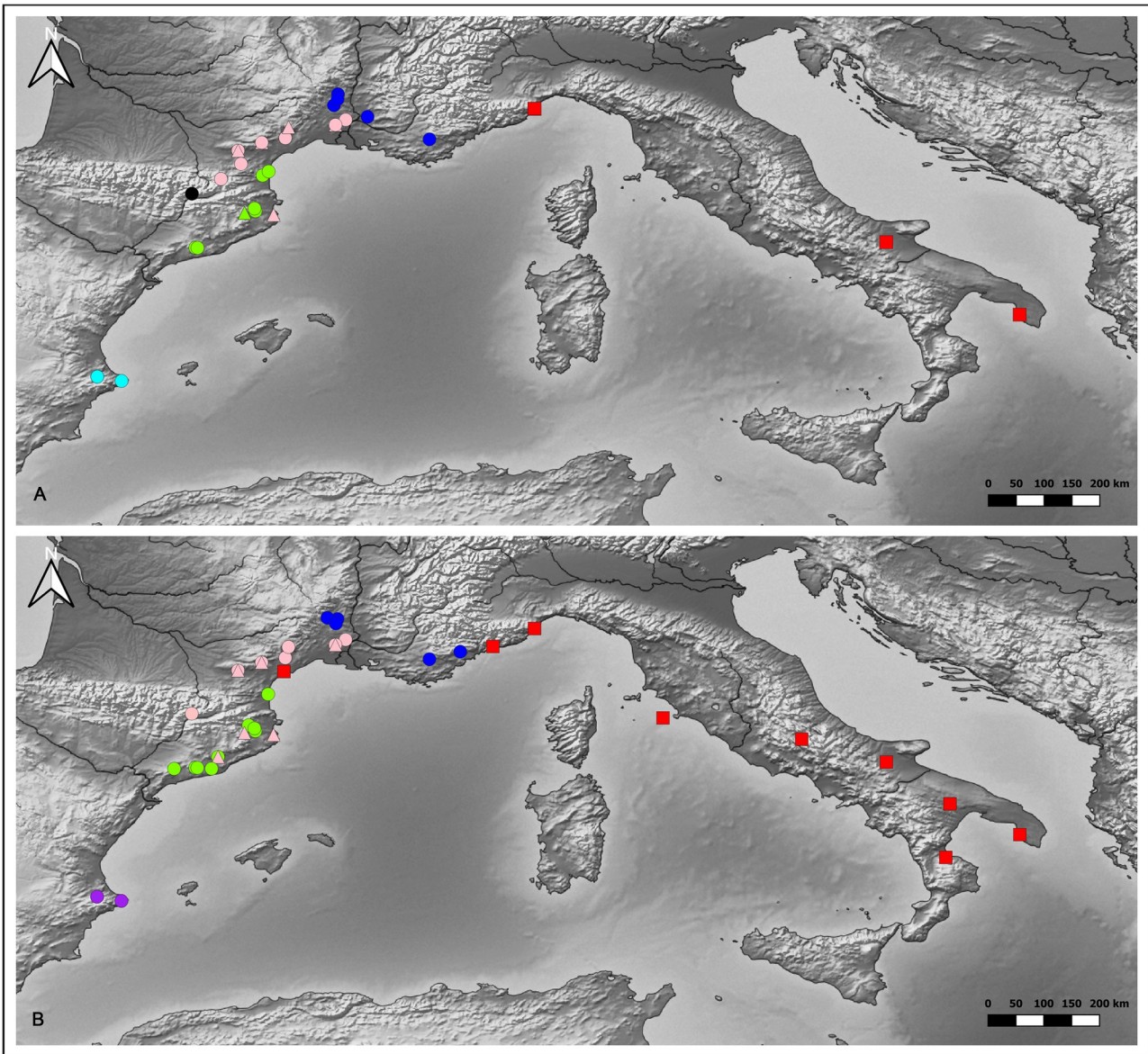

**Fig 1. Location of the archaeological occupations recorded in the database.** (A) archaeological occupations recorded for the pottery decorative techniques. (B) archaeological occupations recorded for the personal ornaments. Marker colours: red–IMP; golden–TyC; blue–RPC; green–LCC; cyan–VC; pink–LCE; purple–VE; black–La Balma Margineda. Marker shapes: square– 1st period; round– 2nd period; triangle– 3rd period. Due to geographical coordinates overlapping, not all archaeological locations markers are visible. Maps were made by D. P. using the software QGIS 3.22.16 [60] and Natural Earth raster maps [61].

eventually the morphology of the decoration itself. Each archaeological assemblage included in the database is described with quantitative variables regarding these decorative attributes.

The personal ornaments dataset includes 88 mutually exclusive bead-types, recorded within 46 occupations and coded as presence/absence in order to avoid any bias due to field methods that may have impacted small items recovery [50,57] (worksheet C, D and H in S1 Dataset). Discrete bead types were created with reference to raw material, morphology, system of suspension (e.g., perforation or groove), size, section, and profile. In the case of animal teeth, we also considered anatomical and species identification.

Geographic coordinates, cultural affiliation, and the corresponding time span of existence (worksheet J in S1 Dataset) are documented for each archaeological occupation of the two datasets. Our sample covers seven archaeological cultures attributed to early farming communities.

## Methods

**Cultural diversity.** Each archaeological occupation was assigned to one of the three main cultural units (Impressa, Cardial, or Epicardial) encompassing the seven Early Neolithic Mediterranean archaeological cultures, as defined in the literature according to lithic technology, settlement patterns, and ceramic production: (Impressa (IMP), Tyrrhenian Cardial (TyC), Rhodanian-Provençal Cardial (RPC), Languedocian-Provençal Cardial (LCC), Valencian Cardial (VC), Languedocian-Provençal Epicardial (LCE), and Valencian Epicardial (VE) (for detailed information see Rigaud et al. 2018, Supplementary Material text A). First, we quantified how the archaeological occupations differed in pottery and personal ornament attributes by using two different similarity indices. The Brainerd-Robinson similarity index [62,63] is appropriate for count data and has been used for calculating pairwise site differences according to their pottery attributes (worksheet A in S2 File). The Jaccard similarity index [64,65] is appropriate for presence/absence data and has been used for calculating pairwise site differences according to their bead-type diversity (worksheet G in S2 File). Both indices are on the scale [0, 1], with a coefficient value of 1 indicating the two assemblages are perfectly similar, while a 0 value indicates that there is no similarity. For more information on the two indexes see Supplementary Text A in S1 File.

**Time-sequential analysis.** In order to explore if, how, and why the network structure may have changed through time we performed a time-sequential analysis. The datasets cover a period of 1,500 years, with secure radiocarbon dates for most of the archaeological occupations or, alternatively, a widely accepted relative chronology [57,66]. We divided the datasets into three separate time sequences: 1) the emergence of farming societies in the region between 8000 BP and 7600 BP years [66]; 2) the period corresponding to an increase in site numbers, a wider spatial distribution of farming occupations in diversified ecosystems, and ceramic style diversification between 7650 BP and 6800 BP years [66]; 3) the end of the Early Neolithic, between 7000 BP and 6500 BP years, corresponding to durable cultural territories and the gradual loss of pottery decoration diversity counterbalanced by an increase in personal ornament diversity [66,67]. This period represents the final phase of the Neolithisation process. We constructed networks for each two consecutive time-sequences, i.e., sequences 1–2 and sequences 2–3.

**Archaeological Similarity Network.** We used Archaeological Similarity Networks (ASN) to explore the degree of connexion between the different archaeological occupations based on their level of similarity [29,68]). In ASN, each node represents an archaeological occupation, the edge drawn between nodes represents the level of similarity between two occupations, and the edge weight (thickness) is a direct reflection of the similarity value [68–70]. In ASN, a higher level of similarity between two archaeological occupations is represented by a thicker edge between the two nodes; conversely, the lower the similarity between two occupations, the thinner the edge.

Due to the expansive chronology considered in this study, many sites were not contemporaneous, even though some belong to the same archaeological culture, and because of the age range inherent to each absolute date, we have no evidence of their strict contemporaneity. Therefore, actual contact between groups is not granted and common cultural traits should mostly be understood as a reflection of transmission through time.

The presence of geographically and chronologically ubiquitous personal ornaments and pottery traits led to a large number of connections, resulting in an overcrowded and dense

network which was barely readable. We therefore imposed a threshold to limit the amount of edges plotted on the network, allowing the underlying network structure to become more apparent and to specifically highlight the node connections of high value [31,32,35]. The threshold chosen was the minimum value that allowed the network to remain as a single component, i.e., the minimum value needed to keep all nodes in the network connected with the exception of a few extreme outliers. Consequently, only the edges with equal or higher value than the threshold are displayed in the drawn networks and, unless otherwise specified, used for all extracted network statistics.

Our network design was performed within R [71] using adapted and extended codes from Brughmans and Peeples [69,72,73] (R protocol in S1 Code). Networks were plotted using two layout methodologies: the Fruchterman-Reingold [74] forced-directed graph layout for a best-fit, and a manual layout using the geographical coordinates of each archaeological occupation.

Different characteristics and information from an archaeological occupation, such as cultural affinity, time-sequence, and geographical area, can be used to inform and help the network's interpretation. For this study, both colour and shape coded each node in the network according to two levels of information: their archaeological culture (colour code), and their time-sequence (symbol code) (worksheets B and D in S1 Dataset).

**Descriptive network statistics.**  Multiple statistics can be calculated to infer on network connectivity, graph structure, and node connections (S1 Table in S1 File). To explore the graph structure and investigate the significance of each node within the network, concerning information retention and transmission, we calculated network density, cluster coefficient, network interval statistics, weighted centralities, and weighted centralization statistics (normalized) for degree, eigenvector, and betweenness centralities [72,73,75–77]. Degree and eigenvector centralities are classical metrics that assess a node's connectivity within a network. Betweenness centrality examines nodes connecting different parts of the network, enabling one to observe how distant segments of the network are interconnected and how information flows between them. The definition and application of each metric is described in S1 Table in S1 File. For each main cultural unit, we also computed its similarity radius, network density, cluster coefficient, and network interval statistics (presented in S4 and S5 Table in S1 File).

**Isolation by distance.**  Understanding how isolation may have impacted cultural flow requires the incorporating of chronological and geographical distances into the analyses of cultural transmission and differentiation [50,78,79]. Both our datasets cover a wide geographical and temporal range. To account for the effect of geographical and chronological distances on cultural diversity, we performed correlation tests between cultural, geographical, and chronological distances using Mantel and partial Mantel tests [80]. Great-circle (spatial) distance was calculated from the latitude and longitude data (R command published by Shennan et al., 2015, worksheets B and H in S2 File). We used the Euclidean distance between the earliest and latest dates for each archaeological culture (worksheets C and I in S2 File). The correlation tests were implemented using Mantel tests (code provided in the S1 Code) and performed using distance matrices that included all the archaeological occupations present in the datasets (full matrices), and the matrices that included only pair of nodes connected in the network built with a threshold (threshold matrices).

## Results

### Pottery decorative techniques

**Archaeological Similarity Networks.**  The pottery decorative techniques networks were built with a threshold of 0.44. The networks (Fig 2) encompass 44 nodes connected by 231 edges, have a density of 0.2442, and a cluster coefficient of 0.7169 (S2 Table in S1 File).

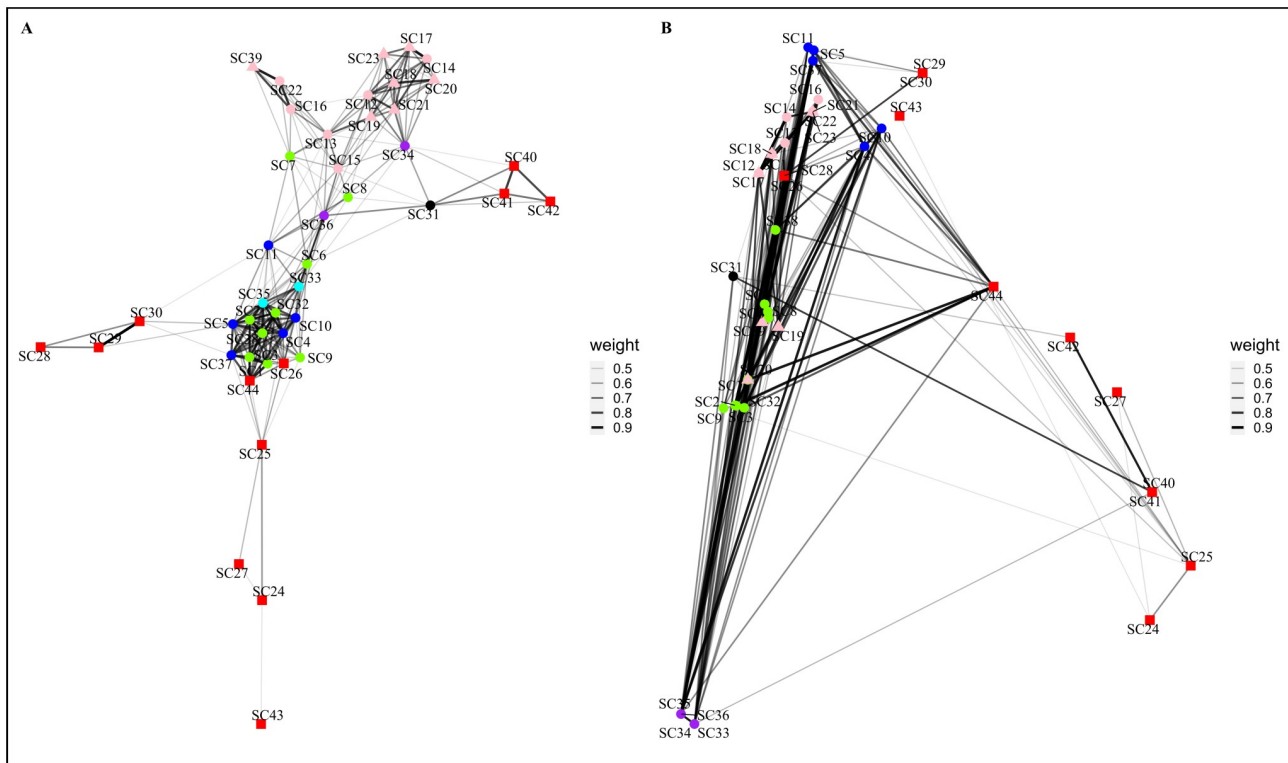

**Fig 2. Archaeological Similarity Network for pottery decorative techniques.** (A) plotted with Fruchterman-Reingold layout. (B) plotted by each occupation's geographical coordinates. ASN's edge weight (thickness) caption to the right of corresponding network. Marker colours: red–IMP; blue–RPC; green–LCC; cyan–VC; pink–LCE; purple–VE; black–La Balma Margineda. Marker shapes: square– 1st period; round– 2nd period; triangle– 3rd period. Due to geographical coordinates overlap, not all archaeological location markers are visible.

The network plotted using the Fruchterman-Reingold layout (Fig 2A) shows two main clusters, each including tightly interconnected nodes. One cluster groups archaeological occupations attributed to the various regional expressions of the Cardial archaeological culture (RPC, LCC and VC), while the other cluster groups archaeological occupations attributed to the different regional expressions of the Epicardial archaeological culture (LCE and VE). Archaeological occupations belonging to the Impressa culture are spread in the peripheral areas of the network, linked to either of the two main clusters. Cardial archaeological occupations are connected by edges showing the highest weight, indicating their high level of similarity. More generally, edges with a high weight mostly connect archaeological occupations belonging to the same archaeological culture.

The network plotted using the geographical coordinates of each archaeological occupation (Fig 2B) shows a smaller number of nodes, due to the various archaeological occupations sharing similar geographical coordinates, documented within the same archaeological site. The network also highlights a geographic gradient with the archaeological occupations attributed to the Impressa, documented within the most eastward sites of the dataset, spread mostly on the right side of the network, and those attributed to the Cardial and the Epicardial, documented within the most westward sites of the dataset, clustering on the left side of the network. The right portion of the network shows few Impressa nodes, connected by few and thin edges. Comparatively, the Cardial and Epicardial occupations, on the left portion of the network, are connected by more numerous and thicker edges.

**Table 1. Pottery decorative techniques and personal ornaments ASN's node weighted centralities and graph centralization scores.**

| | Pottery decorative techniques | | | Personal ornaments | | |
|---|---|---|---|---|---|---|
| | deg | eigen | bet | deg | eigen | bet |
| node level | SC5:12.0034 | SC35: 1.9054 | SC36: 194 | S42: 6.0722 | S29: 2.1549 | S42: 133.333 |
| | SC4: 11.3801 | SC4: 1.8995 | SC25: 120 | S29: 5.7833 | S42: 2.0839 | S34: 109.333 |
| | SC35: 11.1161 | SC32: 1.8886 | SC5: 103 | S40: 5.3409 | S16: 2.0367 | S48: 103.333 |
| | SC32: 10.9880 | SC5: 1.8437 | SC31: 87 | S34: 5.0857 | S41: 2.0367 | S21: 87.333 |
| | SC10: 10.5550 | SC2: 1.8124 | SC34: 79 | S48: 4.8199 | S40: 1.8983 | S40: 84.833 |
| graph level | 0.13663 | 0.18736 | 4.44836E-07 | 0.08402 | 0.21139 | 4.14E-07 |

Only the highest scoring nodes from each node centrality are presented. For a complete node score list, see worksheets D and J in S2 File.

Edge weights (similarity values between pairs of archaeological occupations) for the pottery decorative techniques are distributed within the range of 0.4414 (between occupations SC9-SC37) and 0.9446 (SC29-SC30), with a mean value of 0.6091. The plotted edges range from 0 to 1425 km (SC34-[SC40, SC41]), with a mean geographical distance of 348 km. The chronological distance between the connected occupations ranges from 0 to 775 years (SC34-SC40), with a mean temporal distance of 211 years. All network interval statistics can be found on S3 Table in S1 File.

Archaeological occupations belonging to the Cardial archaeological cultures show the highest degree and eigenvector centrality scores. The highest betweenness centrality scores do not correspond to a specific archaeological culture (Table 1). All graph centralization measures present low values (bellow 0.2). A complete table of node centrality scores is presented in worksheet D in S2 File.

The Cardial cultural unit has both the highest network density and the highest cluster coefficient (0.7333 and 0.8938, respectively), followed by the Epicardial cultural unit (S4 Table in S1 File). These two cultural units correspond to the tightest and most densely organized node clusters. The low values obtained for the Impressa indicate a looser and less dense group of nodes.

The similarity radius wielded similar results when calculated over full and threshold matrices (Table 2). In both cases, the Impressa presented the highest mean similarity radius. The various regional expressions of the Epicardial (LCE and VE), show a higher mean similarity radius than the archaeological cultures belonging to the Cardial unit.

**Isolation by distance.** Correlations calculated between pottery decorative techniques diversity and the spatial and chronological distance matrices show different outcomes if calculated with the full or threshold matrices (Table 3). Using the full matrices, variation in pottery decorative techniques diversity suggests a statistically significant correlation with the geographic distance matrix ($p < 0.05$), with approximately 16.6% of the variance explained by geography. The chronological distance between archaeological sites significantly contributes to approximately 38.7 to 41.6% of the variance. Mantel tests performed on the threshold matrices do not show statistically significant correlations between the pottery attributes or the geographic and chronological distance.

## Personal ornaments

**Archaeological Similarity Network.** The personal ornaments networks were built with a threshold of 0.16. The networks (Fig 3) encompass 46 nodes connected by 217 edges, have a network density of 0.2097, and a cluster coefficient of 0.5470 (S2 Table in S1 File).

**Table 2. Pottery decorative techniques and personal ornaments intra-cultural similarity radius.**

| | Pottery decorative techniques | | | | | |
|---|---|---|---|---|---|---|
| | Full matrices | | | Threshold matrices | | |
| | Max | Mean | Min | Max | Mean | Min |
| IMP | 7930.5250 | 2164.9173 | 0 | 2452.8539 | 776.1145 | 0 |
| RPC | 571.9154 | 218.3470 | 12.8483 | 360.5481 | 195.8913 | 12.8483 |
| LCC | 1101.6831 | 297.2573 | 4.3940 | 373.3214 | 138.2737 | 4.3940 |
| VC | 57.6515 | 57.6515 | 57.6515 | 57.6515 | 57.6515 | 57.6515 |
| LCE | 1465.7943 | 314.2967 | 0 | 680.4047 | 219.9777 | 0 |
| VE | 77.8382 | 77.8382 | 77.8382 | 77.8382 | 77.8382 | 77.8382 |
| | Personal ornaments | | | | | |
| | Full matrices | | | Threshold matrices | | |
| | Max | Mean | Min | Max | Mean | Min |
| IMP | Inf | Inf | 0 | 3856.3917 | 2306.8235 | 0 |
| TyC | 0 | 0 | 0 | 0 | 0 | 0 |
| RPC | Inf | Inf | 44.0010 | 1027.8844 | 390.9746 | 44.0010 |
| LCC | Inf | Inf | 0 | 972.0296 | 294.2533 | 0 |
| VC | 155.407 | 155.407 | 155.407 | 155.4070 | 155.4070 | 155.4070 |
| LCE | Inf | Inf | 0 | 979.8394 | 447.0258 | 0 |
| VE | -Inf | NaN | Inf | -Inf | NA | Inf |

The network plotted using the Fruchterman-Reingold layout (Fig 3A) does not identify any clear cluster, with nodes connected by evenly weighted edges. Archaeological occupations mostly present connections with broadly contemporaneous occupations.

The network plotted using the geographical coordinates of each archaeological occupation (Fig 3B) shows a smaller number of nodes, due to various archaeological occupations sharing similar geographical coordinates, documented within the same archaeological site. The top-left portion of the network shows the majority of nodes and ties. The Impressa archaeological occupations, documented as the most eastward sites of the dataset, are located in the lower-right portion of the network and are loosely connected to the other section of the network. The Valencian archaeological occupations, documented has the most south-westward sites of the dataset, are located in the lower-left portion of the network and are also loosely connected to the other section of the network.

**Table 3. Pottery decorative techniques and personal ornaments Mantel and Partial Mantel tests.**

| | Pottery decor. techn. | | Personal ornaments | |
|---|---|---|---|---|
| Test | Mantel R | P-value | Mantel R | P-value |
| Cultural dist v Geographic dist | 0.165826 | 0.004995 | 0.150148 | 0.003996 |
| Cultural dist v Geographical dist (*t) | 0.121685 | 0.211788 | 0.173578 | 0.128871 |
| Cultural dist vs Geo. dist ~ Chrono. Dist | -0.007375 | 0.545455 | 0.118599 | 0.028971 |
| Cultural dist vs Geo. dist ~ Chrono. dist (*t) | 0.118424 | 0.219780 | 0.179971 | 0.137862 |
| Cultural dist vs Chronological dist | 0.4161334 | 0.000999 | 0.106843 | 0.011988 |
| Cultural dist vs Chronological dist (*t) | 0.028177 | 0.451548 | 0.008415 | 0.505495 |
| Cultural dist vs Chrono. dist ~ Geo. Dist | 0.387084 | 0.000999 | 0.053294 | 0.115884 |
| Cultural dist vs Chrono.dist ~ Geo. dist (*t) | 0.000115 | 0.495504 | -0.049000 | 0.643357 |

(*t) Tests performed over threshold matrices.

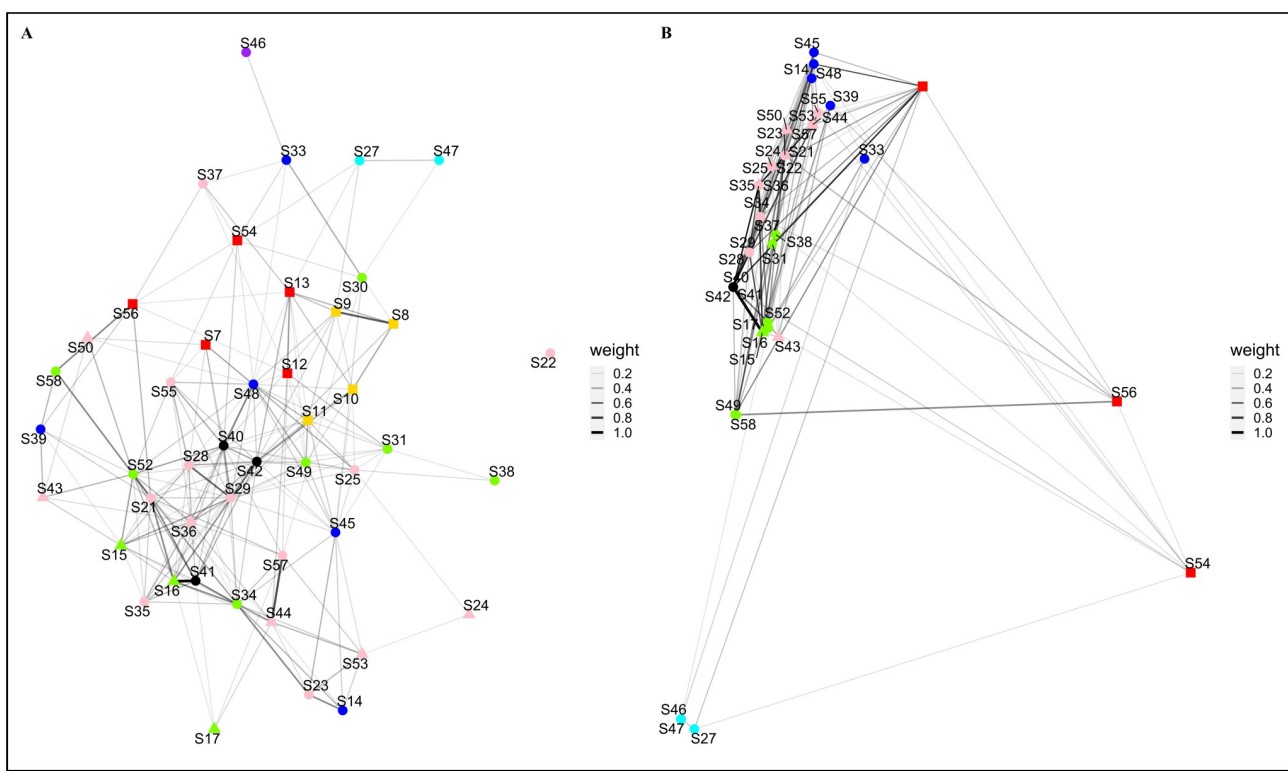

**Fig 3. Archaeological Similarity Network for personal ornaments.** (A) plotted with the Fruchterman-Reingold layout. (B) plotted by each occupation's geographical coordinates. ASN's edge weight (thickness) caption to the right of corresponding network. Marker colours: red–IMP; golden–TyC; blue–RPC; green–LCC; cyan–VC; pink–LCE; purple–VE; black–La Balma Margineda. Marker shapes: square– 1st period; round– 2nd period; triangle– 3rd period. Due to geographical coordinates overlap, not all archaeological location markers are visible.

All archaeological occupation edge weights for the personal ornament ASN's are distributed within the range 0.1613 (between occupations S30-S47) and 1 (S16-S41), with a mean value of 0.2603. Only four edges had a weight above 0.5. The plotted edges range from 0 to 1541 km (S27-S54), with a mean geographical distance of 262 km. Chronological distance between connected occupations range from 0 to 950 years (S37-S56), with a mean temporal distance of 310 years. All network interval statistics can be found in S3 Table in S1 File.

The highest centrality values correspond to archaeological occupations belonging to various archaeological cultures (Table 1). With the exception of eigenvector, all graph centralization measures present low values (bellow 0.2). A complete table of node centrality scores is presented in worksheet J in S2 File.

The Impressa cultural unit has both the highest network density and cluster coefficient (0.4 and 0.6, respectively). The Cardial cultural unit has the second highest network density and the lowest cluster coefficient and the Epicardial shows the lowest network density (S4 Table in S1 File).

The similarity radius for the threshold networks (Table 2) shows that the Impressa presents the highest mean values, and the Languedocian-Catalonian Epicardial has a higher mean similarity radius than the Cardial archaeological cultures. Due to sampling biases, Tyrrhenian Cardial and Valencian Epicardial presented null results: the former because all occupations have the same geographical location, and the latter as it is only represented by a single occupation.

**Isolation by distance.** Correlations calculated between bead-type diversity and the spatial and chronological distance matrices showed different outcomes if calculated with the full or

threshold matrices (Table 3). When using the full matrices, variation in bead-type diversity showed a statistically significant correlation with the geographic distance matrix ($p < 0.05$), with approximately 11.9 to 15% of the variance explained by geography. The chronological distance between archaeological sites significantly contributes to 10.7% of the variance. Mantel tests performed on the threshold matrices do not show statistically significant correlations between personal ornament diversity or geographic and chronological distances.

### Pottery decorative techniques time-sequential analysis

**Archaeological Similarity Networks.**   The pottery decorative techniques sequences 1–2 network (Fig 4A) was built using an imposed threshold of 0.44. The network has 37 nodes connected by 183 edges, a network density of 0.2748, and a cluster coefficient of 0.7302 (S2 Table in S1 File). Archaeological occupations attributed to the various regional expressions of the Cardial almost exclusively cluster together in the central area of the sequences 1–2 network and are connected by heavy weighted edges. Archaeological occupations attributed to the Epicardial are more loosely connected, and the set of Impressa archaeological occupations is fragmented into three separate clusters, loosely connected to the other archaeological occupations.

The pottery decorative techniques sequences 2–3 network (Fig 4B) was built using an imposed threshold of 0.58. The network has 32 nodes connected by 96 edges, a network density of 0.1935, and a cluster coefficient of 0.7830 (S2 Table in S1 File). Archaeological

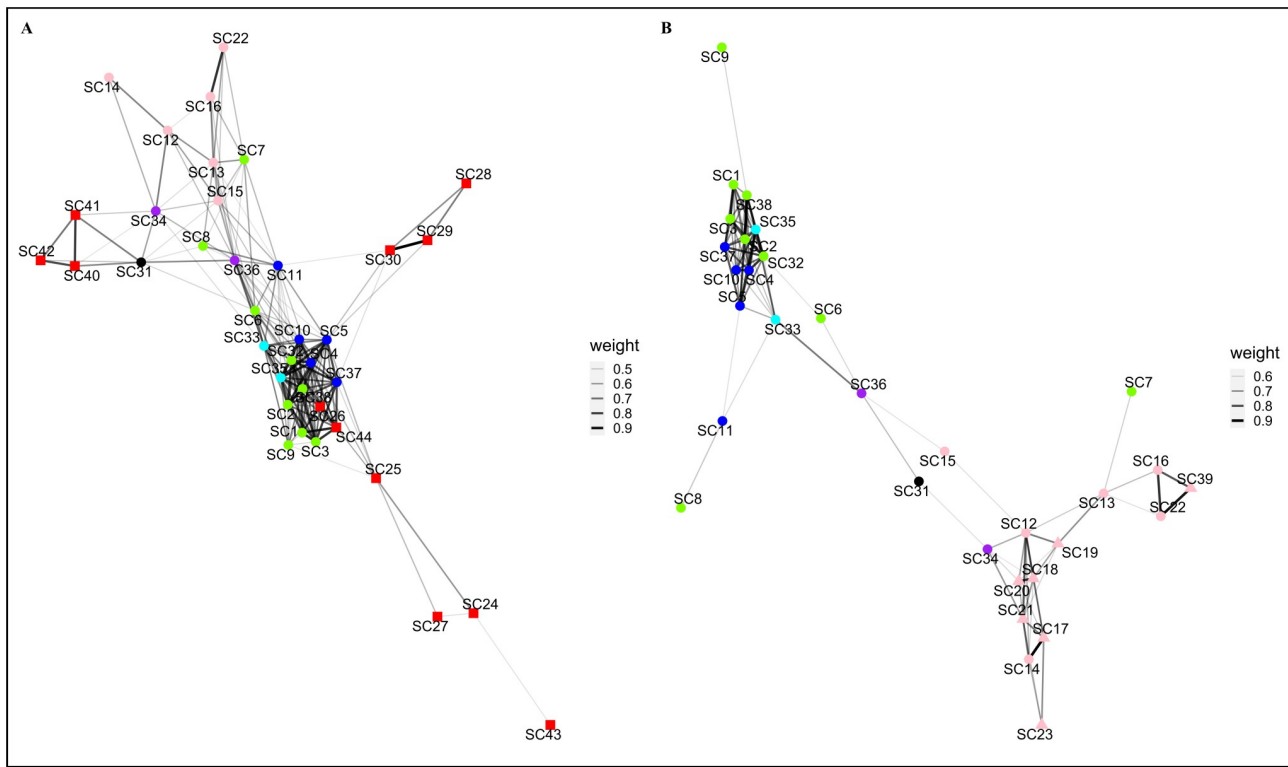

**Fig 4. Archaeological Similarity Network for pottery decorative techniques time-sequential analysis.** (A) sequences 1–2 network. (B) sequences 2–3 network. Plotted with the Fruchterman-Reingold layout. ASN's edge weight (thickness) caption to the right of the corresponding network. Marker colours: red–IMP; blue–RPC; green–LCC; cyan–VC; pink–LCE; purple–VE; black–La Balma Margineda. Marker shapes: square– 1st period; round– 2nd period; triangle– 3rd period. Due to geographical coordinates overlap, not all archaeological locations markers are visible.

occupations are distributed into two loosely connected main clusters linked by a single Valencian Epicardial occupation with low weight edges to each cluster. One of the two main clusters includes almost all of the occupations attributed to the various expressions of the Cardial, while the other cluster includes almost exclusively occupations attributed to the Epicardial. Archaeological occupations are tightly connected within each of the two main clusters. Heavy edges are observed within each of the two clusters, with the Cardial cluster presenting the thicker links.

Archaeological occupation edge weights for the pottery decorative techniques sequences 1–2 ASN are distributed within the range of 0.4414 (between occupations SC9-SC37) and 0.9446 (SC29-SC30), with a mean value of 0.6068. The plotted edges range from 0 to 1425 km (SC34-[SC40, SC41]), with a mean geographical distance of 385 km. Chronological distance between connected occupations ranges from 0 years and 775 years (SC34-SC40), with a mean temporal distance of 210 years. All network interval statistics can be found in S6 Table in S1 File.

Archaeological occupation edge weights for the pottery decorative techniques sequences 2–3 ASN are distributed within the range of 0.5805 (SC9-SC37) and 0.9080 (SC29-SC30), with a mean value of 0.7037. The plotted edges range from 0 (SC12-SC17 & SC21-SC23) to 811 km (SC10-SC35), with a mean geographical distance of 282 km. The chronological distance between occupations ranges from 0 to 500 years (SC12-SC18 & SC31-SC34), with a mean distance of 170 years. All network interval statistics can be found in S6 Table in S1 File.

With a single exception, the highest degree and eigenvector centrality values correspond to archaeological occupations belonging to the various regional expressions of the Cardial (Table 4). The sequences 1–2 network highest betweenness centrality scores belong to various archaeological cultures, while for the sequences 2–3 network the highest values belong, with a single exception, to the Epicardial. All graph centralization measures present low values (below 0.2) for the two pottery decorative techniques time-sequential networks, with the exception of the eigenvector calculated for the sequences 2–3 network. A complete table of node centrality scores is presented in the worksheets E and F in S2 File.

**Table 4. Pottery decorative techniques and personal ornaments time-sequential analysis ASN's node weighted centralities and graph centralization scores.**

| | Pottery decorative techniques | | | | | |
| --- | --- | --- | --- | --- | --- | --- |
| | *TS 1–2* | | | *TS 2–3* | | |
| | degree | eigen | bet | degree | eigen | bet |
| node level | SC5: 12.0034 | SC35: 1.7530 | SC36: 136 | SC32: 8.1966 | SC35: 1.8693 | SC36: 244 |
| | SC4: 11.3800 | SC4: 1.7478 | SC25: 99 | SC38: 7.9980 | SC32: 1.8677 | SC12: 244 |
| | SC35: 11.1161 | SC32: 1.7378 | SC5: 95 | SC35: 7.7032 | SC2: 1.8443 | SC33: 225 |
| | SC32: 10.9880 | SC5: 1.6966 | SC31: 86 | SC2: 7.5677 | SC4: 1.8253 | SC15: 208 |
| | SC10: 10.5550 | SC2: 1.6695 | SC6: 55 | SC4: 7.4861 | SC38: 1.8179 | SC13: 111 |
| graph level | 0.17622 | 0.17338 | 8,68E-07 | 0.13675 | 0.23062 | 1.60E-06 |
| | Personal ornaments | | | | | |
| | *TS 1–2* | | | *TS 2–3* | | |
| | deg | eigen | bet | deg | eigen | bet |
| node level | S42: 5.0722 | S42: 2.3109 | S42: 125.4167 | S29: 5.1167 | S16: 2.0181 | S30: 96.0000 |
| | S29: 4.3833 | S29: 2.2109 | S40: 97.9167 | S34: 5.0857 | S41: 2.0181 | S48: 95.3333 |
| | S40: 4.3052 | S40: 2.0947 | S9: 92.3333 | S42: 4.6167 | S29: 1.9159 | S34: 92.3333 |
| | S48: 3.6381 | S48: 1.6683 | S21: 84.6667 | S16: 4.6000 | S42: 1.7588 | S45: 92.3333 |
| | S21: 3.4024 | S41: 1.5717 | S34: 63.5000 | S41: 4.6000 | S34: 1.7303 | S33: 67.0000 |
| graph level | 0.09639 | 0.26245 | 9.90E-07 | 0.08557 | 0.21597 | 9.78E-07 |

Only the highest scoring nodes for each node centrality are presented. For a complete node score list, see worksheets E, F, K and L in S2 File.

**Isolation by distance.** Correlations calculated between pottery decorative techniques and the spatial and chronological matrices show different outcomes if calculated with the full or threshold matrices (Table 5).

Using the full matrices for the sequences 1–2 network, variation in pottery decorative techniques diversity shows a significant correlation with the geographical matrix (p<0.05), with approximately 18.7% of the variance being explained by geography. The chronological distance between the archaeological occupations significantly contributes to approximately 26.6 to 31.7% of the variance.

Using the full matrices for the sequence 2–3 network, the variation in pottery decorative techniques diversity shows a significant correlation with the chronological matrix, (p<0.05), with approximately 25.2 to 25.3% of the variance being explained by temporal distance.

Mantel tests performed on the threshold matrices of both time-sequential datasets do not show statistically significant correlations between the pottery attributes or the geographic and chronological distances.

### Personal ornaments time-sequential analysis

**Archaeological Similarity Networks.** The personal ornaments sequences 1–2 network (Fig 5A) was built using an imposed threshold of 0.19. The network has 37 nodes connected by 117 edges, a network density of 0.1757, and a cluster coefficient of 0.5134 (S2 Table in S1 File).

The personal ornaments sequences 2–3 network (Fig 5B) was built using an imposed threshold of 0.16 The network has 37 nodes connected by 151 edges, a network density of 0.2267, and a cluster coefficient of 0.6207 (S2 Table in S1 File).

**Table 5. Pottery decorative techniques and personal ornaments time-sequential analysis Mantel and Partial Mantel tests.**

| | Pottery decorative techniques | | | |
| --- | --- | --- | --- | --- |
| | TS 1–2 | | TS 2–3 | |
| Test | Mantel R | P-value | Mantel R | P-value |
| Cultural dist v Geographic dist | 0.187233 | 0.009990 | -0.072005 | 0.921079 |
| Cultural dist v Geographical dist (*t) | 0.111553 | 0.238761 | 0.096484 | 0.364635 |
| Cultural dist vs Geo. dist ~ Chrono. Dist | 0.051056 | 0.227772 | -0.077251 | 0.947053 |
| Cultural dist vs Geo. dist ~ Chrono. dist (*t) | 0.098865 | 0.304695 | 0.107089 | 0.355644 |
| Cultural dist vs Chronological dist | 0.317534 | 0.000999 | 0.251832 | 0.000999 |
| Cultural dist vs Chronological dist (*t) | 0.058371 | 0.370629 | -0.186109 | 0.749251 |
| Cultural dist vs Chrono. dist ~ Geo. Dist | 0.265688 | 0.000999 | 0.253291 | 0.000999 |
| Cultural dist vs Chrono.dist ~ Geo. dist (*t) | 0.026703 | 0.436563 | -0.191678 | 0.778222 |
| | Personal ornaments | | | |
| | TS 1–2 | | TS 2–3 | |
| Test | Mantel R | P-value | Mantel R | P-value |
| Cultural dist v Geographic dist | 0.132137 | 0.026973 | 0.181606 | 0.004995 |
| Cultural dist v Geographical dist (*t) | 0.178398 | 0.239760 | 0.125572 | 0.232767 |
| Cultural dist vs Geo. dist ~ Chrono. Dist | 0.121619 | 0.047952 | 0.182949 | 0.001998 |
| Cultural dist vs Geo. dist ~ Chrono. dist (*t) | 0.145400 | 0.312687 | 0.116618 | 0.274725 |
| Cultural dist vs Chronological dist | 0.052471 | 0.149850 | -0.043506 | 0.726274 |
| Cultural dist vs Chronological dist (*t) | 0.108724 | 0.338661 | -0.142061 | 0.805195 |
| Cultural dist vs Chrono. dist ~ Geo. Dist | -0.006666 | 0.512488 | -0.048969 | 0.765235 |
| Cultural dist vs Chrono.dist ~ Geo. dist (*t) | 0.030257 | 0.464535 | -0.134248 | 0.766234 |

(*t) Tests performed using threshold matrices.

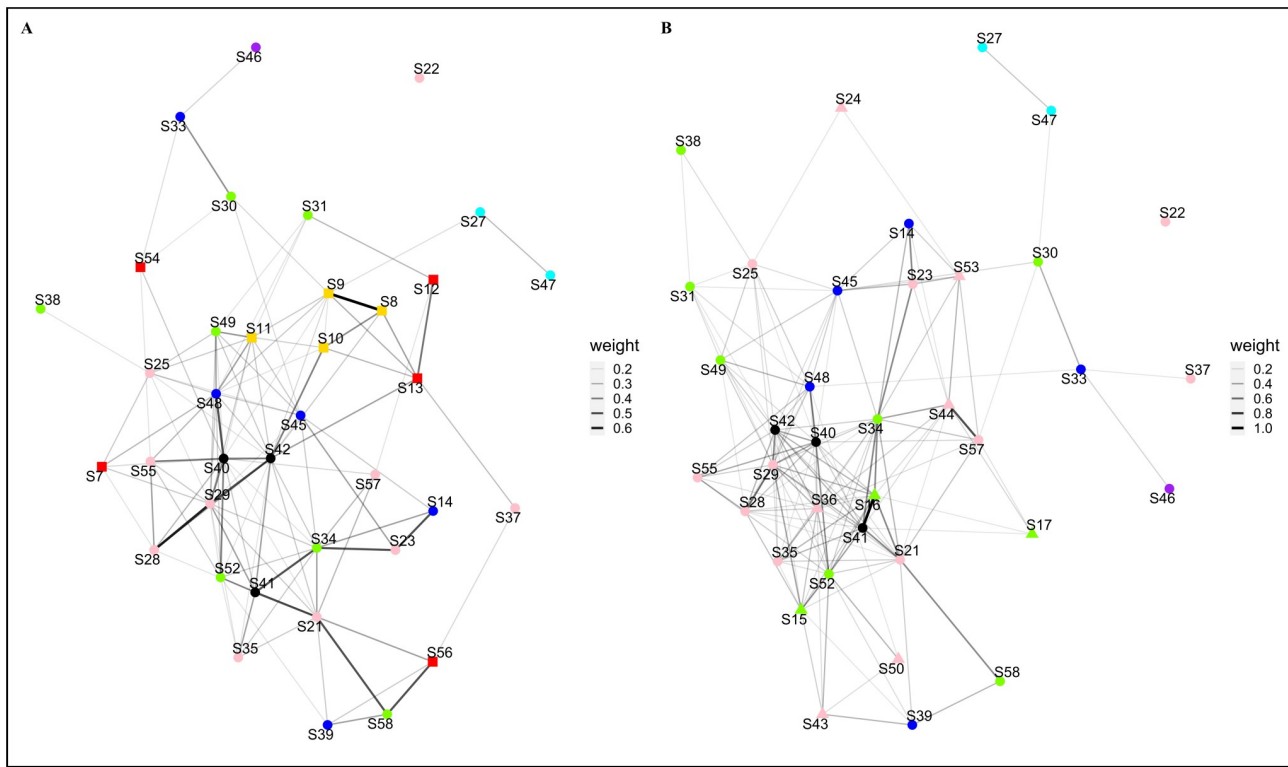

**Fig 5. Archaeological Similarity Network for personal ornaments time-sequential analysis.** (A) sequences 1–2 network. (B) sequences 2–3 network. Plotted with the Fruchterman-Reingold layout. ASN's edge weight (thickness) caption to the right of the corresponding network. Marker colours: red–IMP; golden–TyC; blue–RPC; green–LCC; cyan–VC; pink–LCE; purple–VE; black–La Balma Margineda. Marker shapes: square– 1st period; round– 2nd period; triangle– 3rd period. Due to geographical coordinates overlap, not all archaeological location markers are visible.

The time-sequential networks did not identify any clear cluster, with nodes being connected by evenly weighted edges. Archaeological occupations mostly presented connections with broadly contemporaneous occupations. The archaeological occupations attributed to the Valencian Cardial and Epicardial appear almost isolated in both networks.

Archaeological occupation edge weights for the personal ornaments sequences 1–2 ASN are distributed within the range 0.1905 (between occupations S30-S54) and 0.6667 (S8-S9), with a mean value of 0.2830. The plotted edges range from 0 to 1401 km (S40-S54]), with a mean geographical distance of 294 km. The chronological distance between connected occupations ranges from 0 years and 950 years (S37-S56), with a mean temporal distance of 260 years. All network interval statistics can be found in S6 Table in S1 File.

Archaeological occupation edge weights for the personal ornaments sequences 2–3 ASN are distributed within the range 0.1613 (S30-S47) and 1 (S16-S41), with a mean value of 0.2699. The plotted edges range from 0 to 763 km (S33-S46), with a mean geographical distance of 146 km. The chronological distance between connected occupations range from 0 years to 850 years (S34-[S15, S16]), with a mean temporal distance of 280 years. All network interval statistics can be found in S6 Table in S1 File.

With only two exceptions, the highest personal ornaments time-sequential network centrality values correspond to archaeological occupations from various archaeological cultures attributed to the 2nd period (Table 4). The personal ornaments sequences 2–3 network highest betweenness centrality values correspond to archaeological occupations belonging to various regional Cardial expressions.

For both personal ornaments time-sequential networks, all graph centralization measures presented low values (bellow 0.2), with the exception of eigenvectors. A complete table of node centrality scores is presented in worksheets K and L in S2 File.

**Isolation by distance.** Correlations between personal ornaments diversity and the spatial and chronological distance matrices show different outcomes when considering full or threshold matrices (Table 5).

Using the full matrices for the sequences 1–2 network, variation in personal ornament diversity shows a significant correlation with the geographic distance matrix (p<0.05), with approximately 12.2 to 13.2% of the variance explained by geography.

Using the full matrices for the sequences 2–3 network, variation in personal ornament diversity shows a significant correlation with the geographic distance matrix, (p<0.05), with approximately 18.2 to 18.3% of the variance explained by geography.

Mantel tests performed on the threshold matrices of both time-sequential datasets do not show statistically significant correlations between the personal ornaments or the geographic and chronological distances.

## Discussion

The networks built with pottery decoration techniques data revealed a higher density score, cluster coefficient, and edge weights compared to the ornament network, indicating that the transmission of pottery decorative techniques was characterized by tighter and stronger connections over time and space. Further study of the pottery network shows that the Impressa nodes are loosely connected and separated into three small clusters. Archaeological data and radiocarbon dates available in the region suggest that approximately 8,000 years ago the first small pioneer Neolithic communities sporadically settled in Liguria and Languedoc [81,82]. The loosely connected Impressa nodes may reflect this pioneer, fragmentary, and small-scale peopling event. The low degree of transformation observed during this spatially fragmented pioneer phase is seen as resulting from the process of long-distance maritime displacement [56,83–85] and exploratory behavior [86]. This long-distance connection, highlighted by the higher Impressa mean similarity radius, may reflect these maritime displacements, compared to later periods when the smaller-scale inland dispersions of bigger communities is more likely [83,87]. In contrast, the Cardial archaeological culture nodes are strongly connected and characterized by the highest degree and eigenvector centrality scores, network density, and cluster coefficient, suggesting a significant role in the transmission of pottery decorative techniques over time and space. The lower similarity radius of the Cardial (and inherently smaller geographical distance between connected occupations) show that cultural traits circulated at shorter distances in a denser network of sites. The higher mean edge weights observed in time sequences 2–3 also show that more cultural traits were shared between groups compared to previous time sequences. Absolute chronology evidences a fairly continuous and rapid dispersion of the Neolithic economy during this period (around 7500 cal BP) [19], with the multiplication of sites been seen as a phenomenon of regular demographic expansion [86]. The high level of clustering observed during the Cardial period suggests that it was a time of cultural consolidation, as people settled more densely and permanently over a large territory, fostering intense and regular contacts. Within the following Epicardial, nodes are divided into two main tightly connected clusters, with the Valencian Epicardial in a peripheral position. The post-Cardial network fragmentation (characterized by a lower network density, cluster coefficient, and edge weights calculated for the Epicardial) show changes in the way information flowed between communities, favoring a more regional pattern in the cultural trait circulation. The post-Cardial similarity radius is another indicator of this cultural fragmentation, associated

with large amounts of small-scale contacts. This late network fragmentation episode has been previously documented at a smaller geographic scale in Iberia [51].

The time-sequential analysis confirms this pattern. The Cardial is characterized by many overlapping internal connections with only a few external connections to nodes belonging to other archaeological cultures. The Impressa appears loosely connected to the Cardial in time sequences 1–2 and 2–3; the Epicardial is divided into two portions, barely connected to the Cardial. Mean edge weights slightly increase between the two time-sequential networks indicating that connections become stronger through time as communities spread and densify. The cluster coefficient increases between the two time sequences while the network density decrease. Similarly, the decrease in the mean geographical distance connecting archaeological occupations in sequences 2–3 reflects a regionalization process. The high geographical distance between the LCE occupations is due to the network structure and analysis constrains and is not a contradictory result. Contrary to the Cardial cluster, whose archaeological cultures are all tightly interconnected in the network, the LCE occupations are divided into two groups; however, its network interval statistics are calculated as a single group, which might be the reason for this seemingly dissonant result.

The bead-type similarity network shows a lower cluster coefficient and network density than the pottery decorative technique network, as well as lower mean edge weights. Further analysis suggests that all the nodes appear to be loosely connected with no specific cluster. Despite the increase in both the network density and cluster coefficient between time sequences, no change in the network structure was observed from the Neolithic pioneer phase up to the later phases, corresponding to a demographic boom and Neolithic dispersion [53]. The interval statistics, however, echo results observed for the pottery decorative techniques. The Cardial similarity radius is drastically lower when compared with the previous Impressa period, indicative of the network's densification and regionalization associated to shorter distance connections. The similarity radius in the post-Cardial period also reflects the process of network fragmentation, with the persistence of intense small-scale contacts.

The bead-type similarity network, visibly more open than the pottery network, suggests that the symbolic messages carried by personal ornaments were not particularly impacted by a structured network, compared to pottery decorative techniques, with no particular culture having higher centrality scores than any other. The propensity of personal ornaments to easily circulate in space and time is also evidenced by Mantel tests and interval statistics, which confirm, as previously shown [57], that bead-type associations were spatially more homogeneous than pottery decorative techniques and more persistent over time.

This framework suggests that early farmers' social and symbolic capital was built via complex social networks between groups of people who actively expressed their categorical affiliation through ornaments and pottery decorations.

## Conclusion

Observed differences in the network structures and centralities between pottery and personal ornaments highlight that the two cultural proxies functioned as cultural signals at different social and symbolic levels, and that the transmission of each category of cultural traits was not ruled by similar cultural and social mechanisms [88]. Pottery echoes a regional identity, constantly renegotiated and reinvented by geographical and chronological proximity. Contrastingly, ornaments refer to a common cultural background that developed in large parts of the Mediterranean. Bead-type association contributed to build identities shared at large geographic scales and persisted through time, while pottery styles evolved quickly and were used to constantly reassert more local identities. Relationships built through the

transmission of pottery decorative techniques and bead-types during the transition to farming reveal that the shaping of the social and symbolic capital of the first villagers who settled in the Mediterranean was multifaceted, and probably contributed to multiple successful strategies for developing new social, economic, and cultural opportunities during their dispersion. The adaptation of Near Eastern farming technologies to the European environment [89] most likely resulted from the accumulation and consolidation of new knowledge, facilitated by efficient cooperation and the strong, high-quality social ties that existed among farming communities.

## Supporting information

**S1 Code. Analyses code protocol.**
(TXT)

**S1 Dataset. Archaeological site dataset.** Database of the archaeological sites, layers, variables, and radiocarbon dates used in the analysis.
(XLSX)

**S1 File. Method and results.** Supplementary information for cultural diversity calculation; descriptive statistics definition and application; extra supporting results.
(PDF)

**S2 File. Results.** Cultural similarity, geographical and chronological distance matrices; complete node centralities score tables.
(XLSX)

## Acknowledgments

We would like to acknowledge Alain Queffelec and Mathew Peeples for their help in solving code issues. We thank Jill Cucchi for her help in editing this article.

## Author Contributions

**Conceptualization:** Daniel Pereira, Solange Rigaud.

**Data curation:** Daniel Pereira, Claire Manen, Solange Rigaud.

**Formal analysis:** Daniel Pereira.

**Funding acquisition:** Solange Rigaud.

**Investigation:** Daniel Pereira, Claire Manen, Solange Rigaud.

**Methodology:** Daniel Pereira, Solange Rigaud.

**Project administration:** Solange Rigaud.

**Software:** Daniel Pereira.

**Supervision:** Solange Rigaud.

**Visualization:** Daniel Pereira, Solange Rigaud.

**Writing – original draft:** Daniel Pereira, Solange Rigaud.

**Writing – review & editing:** Daniel Pereira, Claire Manen, Solange Rigaud.

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
