## [Decision Letter · Decision Letter 0]

25 Aug 2023

PONE-D-23-22913The shaping of social and symbolic capital during the transition to farming in the Western Mediterranean: archaeological network analyses of pottery decorations and personal ornamentsPLOS ONE

Dear Dr. Pereira,

Thank you for submitting your manuscript to PLOS ONE. After careful consideration, we feel that it has merit but does not fully meet PLOS ONE’s publication criteria as it currently stands. Therefore, we invite you to submit a revised version of the manuscript that addresses the points raised during the review process. Both reviews are positive. The reviewers offer suggestions for improving your manuscript. Please address these comments while making your revisions. Also, please add an explanation for your use of the three centrality indices. Why were these three (out of the many available) chosen? Each of of the centrality indices measures different network properties. How do the results of each centrality index affect your interpretations of the networks?

We look forward to receiving your revised manuscript.

Kind regards,

John P. Hart, Ph.D.

Academic Editor

PLOS ONE

“This work was supported by the French National Research Agency under the IDEX Bordeaux NETAWA Emergence project № ANR-10-IDEX-03-02 ‘Out of the Core: Exploring social NETworks at the dawn of Agriculture in Western Asia 10 000 years ago’, the CNRS Momentum project ‘Symboling and neighboring at the dawn of Agriculture’ and the Grand Programme de Recherche ‘Human Past’ of the University of Bordeaux.”

“The funders had no role in study design, data collection and analysis, decision to publish, or preparation of the manuscript.

This work was supported by the French National Research Agency under the IDEX Bordeaux NETAWA Emergence project № ANR-10-IDEX-03-02 ‘Out of the Core: Exploring social NETworks at the dawn of Agriculture in Western Asia 10 000 years ago’ [SR], the CNRS Momentum project ‘Symboling and neighboring at the dawn of Agriculture’ [SR] and the Grand Programme de Recherche ‘Human Past’ of the University of Bordeaux [SR].”

Reviewers' comments:

Reviewer's Responses to Questions

**Comments to the Author**

1. Is the manuscript technically sound, and do the data support the conclusions?

Reviewer #1: Yes

Reviewer #2: Yes

2. Has the statistical analysis been performed appropriately and rigorously? 

Reviewer #1: I Don't Know

Reviewer #2: Yes

3. Have the authors made all data underlying the findings in their manuscript fully available?

Reviewer #1: Yes

Reviewer #2: Yes

4. Is the manuscript presented in an intelligible fashion and written in standard English?

Reviewer #1: No

Reviewer #2: Yes

5. Review Comments to the Author

Reviewer #1: This paper addresses important questions regarding the nature of Early Neolithic society in western Europe. It presents the existing literature coherently and appears to be very much up to date (though I am not closely familiar with the details). The datasets appear to be well curated and appropriately structured for network analysis. The choice of similarity measures in the latter is appropriate, and draws on their successful use in recent archaeological network analysis. Clearly the authors are familiar with the latest methods and make extensive use of the latest and best work in the field, that by Brughmans and Peeples in particular.

I find the interpretation of the contrasting patterns observed in pottery vs ornaments quite convincing. Here we have an excellent example of how network analysis can help draw out and visualise patterns in complex datasets that might otherwise be elusive. I do have one observation about the interpretation though, particularly as the authors place emphasis on the identities engendered by bead types (at a wide geographic scale) versus those embodied in ceramic decorative techniques (more localised). In his book ‘Connected Communities’, Peeples makes good use of a sociological distinction between relational and categorical identities. This has also been taken up in a recent review on network science and island archaeology by Helen Dawson. Do the authors think there may be such a distinction in operation here, with beads connected to a kind of categorical identity, while pottery decoration is more linked to relational identity? It seems worth exploring, particularly given the expression of this difference in these recent network approaches in archaeology.

In summary, I feel this will be a great case study for those interested in archaeological network applications, that is to say at a methodological level, as well as contributing to a more nuanced understanding of processes of cultural identity and transmission in the European Early Neolithic. I would accept with some minor revisions – a careful read through by a native speaker, to iron out various infelicities in language, for example, lines 66-68:

correct to “Our ability to store information and make it circulate between individuals and groups is a critical behaviour that signals a tipping point in our evolutionary history”

Line 70, correct to:

How did information and knowledge circulate during the past?

Lines 71-2 also need attention;

Line 79: circulation routes, not roads

Line 124: preventing to explore… correct to ‘hindering an exploration of’, or similar

ETC…

Reviewer #2: Here are some thoughts on the main strengths and areas for improvement in this manuscript:

Strengths:

The study addresses an interesting and important research question about how different types of material culture reflect social networks and knowledge transmission in the past.

The methods are sound, using social network analysis and statistical tests to analyze patterns in two distinct archaeological datasets. The large sample size spanning 1500 years seems appropriate for the addressed topic.

The findings reveal distinct structures and transmission dynamics for pottery vs. personal ornaments, highlighting how each operated differently as cultural proxies. This is a novel contribution.

The discussion links the results back to the broader context of cultural transmission and diffusion during the Neolithic transition.

Areas for improvement:

• The abstract provides a nice overview of the study and highlights the key objectives, methods, and findings, but it is way too long. Some suggestions:

- The first sentence refers to the "storage and circulation of information" - consider rephrasing to be more specific about the types of information being studied here (e.g. symbolic, cultural).

- When introducing the two types of data analyzed (pottery decorations and personal ornaments), it may help to briefly explain why these are useful proxies.

- The summary of findings could highlight more clearly that the two types of data revealed different network structures and transmission patterns (the pottery data showed tighter, more regional networks while the ornament data revealed more widespread, persistent associations). There is no need for a full explanation here.

- The last sentence refers to "social and symbolic capital" - this concept could perhaps be introduced more clearly earlier in the abstract so the connection is evident.

- Make sure the tense is consistent throughout (currently it switches between past and present). Using past tense since this is reporting completed research findings may flow better.

• The introduction could provide more background on why the Neolithic transition is significant and how cultural transmission relates to the research questions.

• The description of the methods/analysis is quite brief - more detail on the network measures, statistical tests, etc. would help readers better evaluate the approach.

• The conclusions could connect back more directly to the original research aims and highlight the wider theoretical implications of the findings.

The structure jumps around a bit - the results section intersperses the main findings with more detailed statistics. Consider reorganizing for clarity.

Careful editing could improve clarity and readability throughout - some sections are dense with archaeological terminology.

Overall the study tackles an important research question and provides novel evidence using social network analysis. However, strengthening the background framing, methods reporting, results presentation, and discussion would improve the clarity and impact of the work. With minor changes, this work can make a valuable contribution to the literature on cultural transmission and Neolithic social dynamics.

6. PLOS authors have the option to publish the peer review history of their article (what does this mean?). If published, this will include your full peer review and any attached files.

Reviewer #1: No

Reviewer #2: No

---

## [Author Response · Author response to Decision Letter 0]

11 Oct 2023

1. We note that Figure 1 in your submission contain [map/satellite] images which may be copyrighted. All PLOS content is published under the Creative Commons Attribution License (CC BY 4.0), which means that the manuscript, images, and Supporting Information files will be freely available online, and any third party is permitted to access, download, copy, distribute, and use these materials in any way, even commercially, with proper attribution. For these reasons, we cannot publish previously copyrighted maps or satellite images created using proprietary data, such as Google software (Google Maps, Street View, and Earth). For more information, see our copyright guidelines: http://journals.plos.org/plosone/s/licenses-and-copyright.

Figure 1 base maps are from Natural Earth (public domain): 

https://www.naturalearthdata.com/about/terms-of-use/

“All versions of Natural Earth raster + vector map data found on this website are in the public domain. You may use the maps in any manner, including modifying the content and design, electronic dissemination, and offset printing. The primary authors, Tom Patterson and Nathaniel Vaughn Kelso, and all other contributors renounce all financial claim to the maps and invites you to use them for personal, educational, and commercial purposes.

No permission is needed to use Natural Earth. Crediting the authors is unnecessary.

However, if you wish to cite the map data, simply use one of the following.

Short text: Made with Natural Earth.

Long text: Made with Natural Earth. Free vector and raster map data @ naturalearthdata.com.”

Editors comment:

Thank you for submitting your manuscript to PLOS ONE. After careful consideration, we feel that it has merit but does not fully meet PLOS ONE’s publication criteria as it currently stands. Therefore, we invite you to submit a revised version of the manuscript that addresses the points raised during the review process.

 Both reviews are positive. The reviewers offer suggestions for improving your manuscript. Please address these comments while making your revisions. Also, please add an explanation for your use of the three centrality indices. Why were these three (out of the many available) chosen? Each of the centrality indices measures different network properties. How do the results of each centrality index affect your interpretations of the networks?

R: We would like to express our gratitude to the editor for handling our manuscript and for providing valuable comments. In our study, we have employed three network centralities, namely degree, eigenvector, and betweenness centralities, measured both at the node and graph levels. Our primary objective was to investigate the significance of each node within the network concerning information retention and transmission.

Two fundamental ways to assess these aspects of network information flow are by examining the level of connectivity of each node within the network and their positioning in critical points of the transmission chain. Degree and eigenvector centralities are classical metrics that assess a node's connectivity within a network. Eigenvector centrality, in particular, not only identifies highly connected nodes but also highlights those with the highest social importance, as it accounts for connections to highly connected nodes. These two metrics effectively address both facets of information flow mentioned above.

However, given the nature of our networks, which exhibit various distinct groups or clusters, we have chosen to include a third metric: betweenness centrality. The incorporation of betweenness centrality, a metric that examines nodes connecting different parts of the system, enables us to explore how distant and sometimes diverse segments of the network are interconnected and how information traverses between them.

To enhance the clarity of our manuscript for readers, we have included a brief explanation in the Methods section that outlines the rationale behind our selection of each centrality metric.

Reviewer #1 comments:

This paper addresses important questions regarding the nature of Early Neolithic society in western Europe. It presents the existing literature coherently and appears to be very much up to date (though I am not closely familiar with the details). The datasets appear to be well curated and appropriately structured for network analysis. The choice of similarity measures in the latter is appropriate, and draws on their successful use in recent archaeological network analysis. Clearly the authors are familiar with the latest methods and make extensive use of the latest and best work in the field, that by Brughmans and Peeples in particular.

I find the interpretation of the contrasting patterns observed in pottery vs ornaments quite convincing. Here we have an excellent example of how network analysis can help draw out and visualise patterns in complex datasets that might otherwise be elusive. I do have one observation about the interpretation though, particularly as the authors place emphasis on the identities engendered by bead types (at a wide geographic scale) versus those embodied in ceramic decorative techniques (more localised). In his book ‘Connected Communities’, Peeples makes good use of a sociological distinction between relational and categorical identities. This has also been taken up in a recent review on network science and island archaeology by Helen Dawson. Do the authors think there may be such a distinction in operation here, with beads connected to a kind of categorical identity, while pottery decoration is more linked to relational identity? It seems worth exploring, particularly given the expression of this difference in these recent network approaches in archaeology.

R: We would like to thank the reviewer for the time taken to review our manuscript and for the comments and suggestions made for its improvement. Because our dataset relies exclusively on symbolic productions—pottery decorations and personal ornaments—our results primarily reflect categorical identities. If we had also considered the compositional data of pottery, we might have been able to uncover relational identities, as suggested by Peeple in his work titled 'Identity and Social Transformation 

in the Prehispanic Cibola World: A.D. 1150-1325.' Furthermore, we believe that personal ornaments could potentially provide insights into relational identities at a smaller scale. However, achieving this level of information would require more refined contextual and chronological data, potentially including primary deposits such as burials, in order to extend our analysis to the intra-group social composition. However, we agree that categorical identity is a key notion in our work and we briefly introduce it at the end of the discussion in the new version of the manuscript. We thank the reviewer and we will likely dig deeper this topic in the future. 

In summary, I feel this will be a great case study for those interested in archaeological network applications, that is to say at a methodological level, as well as contributing to a more nuanced understanding of processes of cultural identity and transmission in the European Early Neolithic. I would accept with some minor revisions – a careful read through by a native speaker, to iron out various infelicities in language, for example, lines 66-68:

correct to “Our ability to store information and make it circulate between individuals and groups is a critical behaviour that signals a tipping point in our evolutionary history”

Line 70, correct to:

How did information and knowledge circulate during the past?

Lines 71-2 also need attention;

Line 79: circulation routes, not roads

Line 124: preventing to explore… correct to ‘hindering an exploration of’, or similar

ETC…

R: We thank the reviewer for pointing out these problems. We have reviewed the manuscript and made the suitable changes. We have also submitted our revised manuscript for proofing by a native English speaker. We hope the changes addressed helped in making our revised manuscript clearer.

Reviewer #2 comments:

Here are some thoughts on the main strengths and areas for improvement in this manuscript:

Strengths:

The study addresses an interesting and important research question about how different types of material culture reflect social networks and knowledge transmission in the past.

The methods are sound, using social network analysis and statistical tests to analyze patterns in two distinct archaeological datasets. The large sample size spanning 1500 years seems appropriate for the addressed topic.

The findings reveal distinct structures and transmission dynamics for pottery vs. personal ornaments, highlighting how each operated differently as cultural proxies. This is a novel contribution.

The discussion links the results back to the broader context of cultural transmission and diffusion during the Neolithic transition.

Areas for improvement:

• The abstract provides a nice overview of the study and highlights the key objectives, methods, and 

findings, but it is way too long. Some suggestions:

- The first sentence refers to the "storage and circulation of information" - consider rephrasing to be more 

specific about the types of information being studied here (e.g. symbolic, cultural).

- When introducing the two types of data analyzed (pottery decorations and personal ornaments), it may help to briefly explain why these are useful proxies.

- The summary of findings could highlight more clearly that the two types of data revealed different network structures and transmission patterns (the pottery data showed tighter, more regional networks while the ornament data revealed more widespread, persistent associations). There is no need for a full explanation here.

- The last sentence refers to "social and symbolic capital" - this concept could perhaps be introduced more clearly earlier in the abstract so the connection is evident.

- Make sure the tense is consistent throughout (currently it switches between past and present). Using past tense since this is reporting completed research findings may flow better.

R: We would like to thank the reviewer for the time taken to review our manuscript and for the various comments and suggestions made to improve its quality. We have reviewed the abstract to make it more concise, clearer and keep the tense consistent following his advices.

• The introduction could provide more background on why the Neolithic transition is significant and how cultural transmission relates to the research questions.

R: We have detailed in the introduction that the Neolithic revolution played a pivotal role in establishing the economic and social underpinnings upon which many present-day societies rely. This includes diverse technics for food production and storage, the emergence of surpluses, the shift towards sedentism, the specialization of labor, the growth of social complexity, and, ultimately, the formation of state institutions.

We also added that previous studies have shown that circulation networks favoured both cultural and demic diffusions and that each mechanisms acted at different extent in the various region of Europe (Fort 2012, Fort 2022).

• The description of the methods/analysis is quite brief - more detail on the network measures, statistical tests, etc. would help readers better evaluate the approach.

R: In our methods section we tried to give a clear and detailed description of our network construction method, including threshold level decision process, and the detailing of other tests performed along with the reasoning for such. For the sake of clarity and brevity, where we saw this would not hinder a readers understanding of the methods, we decided to leave the description/ definition of some of the metrics, for the supplementary materials. In Supplementary Text A (S1 Supplementary Materials) we further clarify our choice of similarity indices. In Supplementary Table 1 (S1 Supplementary Materials) we present for each of the various metrics used, a definition and application. We feel that including these information in the text would make the methods section saturated and cumbersome for the reader, while its presentation 

in the supplementary materials is a clearer and viable form. In order to improve our manuscript and make it clearer to the readers, we included in the Methods section of the main text a short explanation on the reasons for each metric choice, and indicated in a clearer way that further details can be found in the S1 Supplementary Information’s document.

• The conclusions could connect back more directly to the original research aims and highlight the wider theoretical implications of the findings.

R: We have added to the conclusion a short sentence that recalls the very first themes introduced at the very beginning of the paper, by stating that cooperation, accumulation and consolidation of knowledge’s likely allowed communities to adapt near eastern farming technologies to the European environment.

The structure jumps around a bit - the results section intersperses the main findings with more detailed statistics. Consider reorganizing for clarity.

Careful editing could improve clarity and readability throughout - some sections are dense with archaeological terminology.

R: The deep editing of the text by a profession English editor has allowed to fix those issues.

Overall the study tackles an important research question and provides novel evidence using social network analysis. However, strengthening the background framing, methods reporting, results presentation, and discussion would improve the clarity and impact of the work. With minor changes, this work can make a valuable contribution to the literature on cultural transmission and Neolithic social dynamics.

---

## [Editor Report · Decision Letter 1]

25 Oct 2023

The shaping of social and symbolic capital during the transition to farming in the Western Mediterranean: archaeological network analyses of pottery decorations and personal ornaments

PONE-D-23-22913R1

Dear Dr. Pereira,

We’re pleased to inform you that your manuscript has been judged scientifically suitable for publication and will be formally accepted for publication once it meets all outstanding technical requirements.

Kind regards,

John P. Hart, Ph.D.

Academic Editor

PLOS ONE
---

## [Editor Report · Acceptance letter]

31 Oct 2023

PONE-D-23-22913R1 

The shaping of social and symbolic capital during the transition to farming in the Western Mediterranean: archaeological network analyses of pottery decorations and personal ornaments 

Dear Dr. Pereira:

I'm pleased to inform you that your manuscript has been deemed suitable for publication in PLOS ONE. Congratulations! Your manuscript is now with our production department. 

Kind regards, 

on behalf of

Dr. John P. Hart 

Academic Editor

PLOS ONE